# Screening and Identification of a Prognostic Model of Ovarian Cancer by Combination of Transcriptomic and Proteomic Data

**DOI:** 10.3390/biom13040685

**Published:** 2023-04-18

**Authors:** Jinghang Jiang, Zhongyuan Chen, Honghong Wang, Yifu Wang, Jie Zheng, Yi Guo, Yonghua Jiang, Zengnan Mo

**Affiliations:** 1Center for Genomic and Personalized Medicine, Guangxi Medical University, Nanning 530021, China; 2Guangxi Collaborative Innovation Center for Genomic and Personalized Medicine, Nanning 530021, China; 3Guangxi Key Laboratory for Genomic and Personalized Medicine, Guangxi Medical University, Nanning 530021, China; 4Graduate School, Guangxi Medical University, Nanning 530021, China; 5Collaborative Innovation Centre of Regenerative Medicine and Medical BioResource Development and Application Co-constructed by the Province and Ministry, Guangxi Medical University, Nanning 530021, China; 6School of Public Health, Guangxi Medical University, Nanning 530021, China

**Keywords:** ovarian cancer, proteome, transcriptome, tumor-infiltrating immune cells, single cell

## Abstract

The integration of transcriptome and proteome analysis can lead to the discovery of a myriad of biological insights into ovarian cancer. Proteome, clinical, and transcriptome data about ovarian cancer were downloaded from TCGA’s database. A LASSO–Cox regression was used to uncover prognostic-related proteins and develop a new protein prognostic signature for patients with ovarian cancer to predict their prognosis. Patients were brought together in subgroups using a consensus clustering analysis of prognostic-related proteins. To further investigate the role of proteins and protein-coding genes in ovarian cancer, additional analyses were performed using multiple online databases (HPA, Sangerbox, TIMER, cBioPortal, TISCH, and CancerSEA). The final resulting prognosis factors consisted of seven protective factors (P38MAPK, RAB11, FOXO3A, AR, BETACATENIN, Sox2, and IGFRb) and two risk factors (AKT_pS473 and ERCC5), which can be used to construct a prognosis-related protein model. A significant difference in overall survival (OS), disease-free interval (DFI), disease-specific survival (DSS), and progression-free interval (PFI) curves were found in the training, testing, and whole sets when analyzing the protein-based risk score (*p* < 0.05). We also illustrated a wide range of functions, immune checkpoints, and tumor-infiltrating immune cells in prognosis-related protein signatures. Additionally, the protein-coding genes were significantly correlated with each other. EMTAB8107 and GSE154600 single-cell data revealed that the genes were highly expressed. Furthermore, the genes were related to tumor functional states (angiogenesis, invasion, and quiescence). We reported and validated a survivability prediction model for ovarian cancer based on prognostic-related protein signatures. A strong correlation was found between the signatures, tumor-infiltrating immune cells, and immune checkpoints. The protein-coding genes were highly expressed in single-cell RNA and bulk RNA sequencing, correlating with both each other and tumor functional states.

## 1. Introduction

Ovarian cancer is one of the deadliest gynecological malignancies in the world [1,2,3]. According to Global Cancer Statistics 2020, 313,959 confirmed cases of ovarian cancer were reported, and 207,252 deaths were the result of ovarian cancer in 2020 [4]. There is a rapid cancer profile transition occurring in China alongside a significant increase in the incidence of cancer [5]. From 1990 to 2019, the net drift rate of ovarian cancer as a cause of death in China was 0.76% (95% CI 0.57–0.95) per year [6]. A lack of visible clinical manifestations during the early stages of the disease [1,5,7] means that ovarian cancer is often not detected early with non-invasive methods, and thus most individuals who are diagnosed with the disease are already in the mid to late stages. As there are few treatment options that can improve the prognosis of ovarian cancer patients, most patients eventually die from the disease due to disease recurrence and metastasis.

It would be incredibly useful for ovarian cancer management to have biomarkers that could reliably predict a patient’s outcome [7,8]. As a heterogeneous disease, ovarian cancer contains several distinct subtypes that involve heterogeneous populations of cells and proteins [2,9]. The research focuses on the analysis of phenomes, genomics, transcriptomes, and proteomes, among other biomolecules, in various diseases, including ovarian cancer. Moreover, proteomics analyses can complement transcriptomic and genomic analyses, providing insights into the underlying mechanisms and potentially serving as objective biomarkers of disease. Scholars have gained a better understanding of the functions of proteins in the proteome of ovarian cancer and their impact on prognosis, providing new avenues for developing more effective diagnostics and therapies [10,11]. In the process of tumor development and metastasis, communication between tumor cells and their microenvironment is critical. Some scholars have investigated tumor-infiltrating lymphocytes (TILs) and their diverse biological processes in the tumor microenvironment (TME), providing insights into the immune responses elicited by ovarian cancer [3,12,13]. As a result of modern cancer research, precision proteomic strategies are increasingly being considered to identify prognostic and predictive biomarkers and guide therapy selection [14,15,16]. In addition, the TME and proteomics have been studied in the context of the initiation and progression of ovarian cancer. Therefore, identifying useful prognosis-related proteins of ovarian cancer would be a tremendous help for managing ovarian cancer treatment.

It is necessary to integrate and validate these results by combining them with proteomic and transcriptomic data to obtain a comprehensive understanding of how the genome affects protein activity [17]. Proteome profiling in The Cancer Genome Atlas Database (TCGA) is a powerful proteomic approach to understanding signaling pathways and tumor development. Therefore, proteome and transcriptome profiling have been integrated to identify markers of ovarian cancer that may be associated with prognosis and develop new therapeutic targets. We developed a model that utilizes proteins associated with ovarian cancer, validated its clinical usefulness for prognosis with individual predictive capability, and explored the interaction between these proteins and the TME. The proteins and protein-coding genes were validated through multidimensional analysis using online databases (HPA, Sangerbox, TIMER, cBioPortal, TISCH, and CancerSEA).

## 2. Materials and Methods

### 2.1. Extraction of Data

Proteome profiling data on ovarian cancer were downloaded from TCGA database (https://portal.gdc.cancer.gov/, accessed on 8 August 2021) [18]. For the acquisition of corresponding clinical and transcriptome data of matched ovarian cancer individuals, we used TCGA database. Next, we combined the protein data with clinical and transcriptome data using information derived from the ID number through Perl software and R’s (https://www.r-project.org/, accessed on 8 August 2021) “impute” package. Patients with missing protein data and incomplete clinical information were excluded, and 417 patients’ data were enrolled in the final analysis.

### 2.2. Construction and Assessment of a Prognosis-Related Protein Model

To ensure that there was no overlap between the training (*n* = 209) and testing (*n* = 208) sets, all ovarian cancer patients were randomly assigned to two groups in a 1:1 ratio using “caret” package. We utilized logistic least absolute shrinkage and selection operator (LASSO) regressions on the training set to select proteins that were independently associated with prognosis risk. Then, we assessed the prediction performance using testing and whole sets. An evaluation of prognostic factors was conducted via univariate Cox regression and Kaplan–Meier survival analysis with *p* < 0.05 as a fiducial point and using the “survival” package in the training set. A volcano plot was visualized using the R packages “dplyr”, “ggplot2”, and “ggrepel”. Then, protein biomarkers were input into the subsequent LASSO to identify the non-zero coefficient proteins using the “glmnet” package. A multivariate Cox regression analysis of the LASSO data was performed using “survival” package to identify proteins related to prognosis. To graphically demonstrate the effects of the predictors, a principal component analysis (PCA) was carried out. Eventually, we identified proteins with significant associations with prognosis, either as protective (hazard ratio HR > 1 and *p* < 0.05) or as risk proteins (HR < 1 and *p* < 0.05). Each patient’s risk score was calculated by adding the proteins’ expression levels by their regression coefficients (β) for all selected proteins. Protein risk scores = ∑^n^_i = 1_β_i_ ∗ expression of protein_i_. Using the median integrated risk score, high-risk and low-risk groups were differentiated. Using Kaplan–Meier analysis, survival data were visualized, and the log-rank test was utilized to identify any significant differences in overall survival (OS), disease-free interval (DFI), disease-specific survival (DSS), and progression-free interval (PFI) between groups. The “survival” and “survminer” packages were used for the analysis across the training, testing, and whole sets. An R package called pheatmap was used to draw the risk score, survival time, and protein expression between the high-risk and low-risk groups across the training, testing, and whole sets. In order to determine the predictive accuracy of the protein risk score, we measured the specificity, sensitivity, and area under the receiver operating characteristic (ROC) curve. The “survival”, “survminer”, and “timeROC” packages were used to generate the ROC curves and AUC values.

### 2.3. Verification of Clinical Relevance and Prognostic Protein Expression

We extracted survival data from clinical records and excluded those with incomplete details of clinicopathological characteristics, survival time, and survival status. Ultimately, patients with OS, DFI, DSS, and PFI data were included in the survival analysis. A median risk score and the expression of individual prognostic-related protein signatures were used to distinguish high-risk and low-risk patients. To measure the survival difference between the two groups, a Kaplan–Meier survival analysis was performed using the survival R package and the survminer R package. Next, the ggpubr R package was used for analysis to explore the potential relationship between the risk score and clinical variables. A threshold value was set for all proteins. Protein model-associated proteins were filtered and calculated with a *p*-value threshold of 0.001 and correlation coefficients of 0.40. The R package corrplot, as well as ggalluvial, was used to generate circle and Sankey diagrams.

### 2.4. Construction of a Nomogram for Clinical Survival Prediction

We developed a nomograph that can accurately predict the survival of ovarian cancer patients, thus enhancing the practicability of our prognostic model. The risk score and other clinicopathological variables were used to conduct univariate and multivariate Cox proportional hazard analyses, and a nomogram was then constructed based on the results. Additionally, calibration curves were plotted to evaluate the accuracy of the nomogram. The R packages “survival”, “regplot”, and “rms” were used.

### 2.5. Consensus Clustering Analysis of Prognostic-Related Proteins

For the consensus molecular subtyping of prognostic-related protein subtypes, consensus clustering analysis was performed via the “ConsensusClusterPlus” R package. The consensus clustering method can identify unsupervised classes in datasets. Additionally, Kaplan–Meier curves were created, and log-rank tests were carried out on the different groups to examine the differences in OS, DFI, DSS, and PFI.

### 2.6. Gene Set Enrichment Analysis (GSEA) and Single-Sample GSEA (ssGSEA)

To further confirm the biological characteristics and pathways related to the prognostic-related protein risk scores, GSEA was implemented using the R packages “clusterProfiler” and “enrichplot”, as well as the “c2.cp.kegg.symbols.gmt” gene sets obtained from the Molecular Signatures Database (MSigDB). With the R packages “reshape2”, “ggpubr”, “GSEABase”, and “GSVA”, a ssGSEA enrichment score represents the relative abundance of immune cell types. Analyses of immunogenomics reveal protein-immunophenotype relationships and predictors of responses to checkpoint blockades.

### 2.7. An Analysis of Proteins and Protein-Coding Genes Using Online Databases

An analysis of the expression of prognostic-related proteins in normal and carcinoma tissues was performed using The Human Protein Atlas (HPA) (www.proteinatlas.org, accessed on 8 August 2021) [19]. Using data from TCGA and GTEx databases, Sangerbox (http://vip.sangerbox.com/, accessed on 8 August 2021) [20] was used to analyze protein-coding gene expression. After that, we used TIMER (https://cistrome.shinyapps.io/timer/, accessed on 8 August 2021) to determine the correlations between protein-coding genes [21]. The cBioPortal (http://cbioportal.org, accessed on 8 August 2021) [22] was used to assess the mutation status of protein-coding genes in TCGA and ICGC datasets. As part of our study, we examined the EMTAB8107 and GSE154600 single-cell datasets of ovarian cancer to understand the expression of protein-coding genes among different cell clusters in the TISCH database (http://tisch.comp-genomics.org/, accessed on 8 August 2021) [23]. Lastly, to study the functional status of cancer cells at the level of individual cells, CancerSEA was developed to assess protein-coding genes (http://biocc.hrbmu.edu.cn/CancerSEA/, accessed on 8 August 2021) [24].

### 2.8. Statistical Analysis

Data analysis was performed using R software 4.1.0 (https://www.r-project.org/, accessed on 8 August 2021) and Perl tools (https://www.perl.org, accessed on 8 August 2021). Prognostic factors were analyzed using univariate regression, LASSO regression, and multivariate Cox regression. Survival over time could be assessed via a Kaplan–Meier analysis. The R packages “dplyr”, “ggplot2”, “glmnet”, “survminer”, “pheatmap”, “survival”, “survminer”, “timeROC”, “reshape2”, “ggpubr”, “GSEABase”, and “GSVA” were used to run this analysis. Statistical significance was defined as *p* ≤ 0.05.

## 3. Results

### 3.1. Proteins with Prognostic Relevance in Ovarian Cancer

As a result of univariate Cox regression analyses, among the prognostic factors, 30 were found to be differentially expressed, with 18 having a suppressive effect (hazard ratio HR < 1) and 12 having a promotive effect (HR > 1). Proteins whose expressions were differentially expressed were plotted on a volcano plot (Figure 1A). A simultaneous analysis of the forest plots of the prognostic factors of HR and the 95% confidence intervals of the risk rates for every single protein is presented in Figure 1B. A total of 30 protein biomarkers were used in the subsequent LASSO regression correlation analysis of ovarian cancer (Figure 1C,D). The final resulting prognosis factors consisted of seven protective factors (P38MAPK, RAB11, FOXO3A, AR, BETACATENIN, Sox2, and IGFRb) and two risk factors (AKT_pS473 and ERCC5). Using these nine proteins, a PCA was performed for all ovarian cancer patients, and the two groups were easily distinguished (Figure 1E,F).

### 3.2. Construction and Assessment of Prognosis-Related Protein Model

In order to predict the prognosis of ovarian cancer patients, the prognostic-related protein risk score for differentially expressed proteins was calculated for each ovarian cancer tissue. The prognostic-related protein risk score = 0.328 × AKT_pS473 + 0.442 × ERCC5 − 0.399 × Sox2 − 0.495 × AR − 0.419 × BETACATENIN − 0.318 × IGFRb − 0.826 × FOXO3A − 1.5743 × P38MAPK − 1.570 × RAB11. A median risk score was used to categorize patients in the training set into high-risk (*n* = 105) and low-risk groups (*n* = 104). A robust difference in the OS curves was established between the high- and low-risk groups (*p* < 0.001) (Figure 2A). An increase in the mortality of the training set with an increase in the risk score can be seen in Figure 2B,C. The prognosis factors are presented visually with heat maps (Figure 2D). Based on the area under the ROC curve, the predicted survival rates of ovarian cancer were 0.722 at 1 year, 0.731 at 3 years, and 0.749 at 5 years with an accuracy in the training set (Figure 2E). Using the testing set (114 cases with high scores and 94 cases with low scores) and the whole set (218 cases with high scores and 199 cases with low scores), we confirmed the accuracy of our protein signature. According to the Kaplan–Meier analyses, patients with high protein risk scores had shorter OS in the testing and whole sets, as can be observed in Figure 2F–K. We also determined the protein risk score, survival status, and protein expression signature of the testing (Figure 2G–I) and whole (Figure 2L–M) sets of patients with ovarian cancer. According to the study, ROC analysis was performed, and the AUC values for 1-year, 3-year, and 5-year survival were 0.664, 0.596, and 0.644 in the testing set and 0.697, 0.665, and 0.694, in the whole sets, respectively (Figure 2J,O).

As a next step, we examined the DFI, DSS, and PFI prognostic significance of protein risk scores in the ovarian cancer cohort. As the DFI was determined by the protein risk score, 39 patients were classified as high risk and 59 as low risk in the training set, 56 patients were classified as high risk and 53 as low risk in the testing set, and 95 patients were classified as high risk and 112 as low risk in the whole set. The patient DFI tended to be significantly longer in patients with low risk than in patients with high risk in the training set (*p* < 0.001) (Figure 3A), testing set (*p* = 0.016) (Figure 3B), and the whole set (*p* < 0.001) (Figure 3C). Based on ROC curve analysis, the AUC for 3- and 5-year survival was 0.756 and 0.737; 0.557 and 0.655; and 0.654 and 0.693 in the training (Figure 3D), testing (Figure 3E) and whole (Figure 3F) sets, respectively. In accordance with the scores, patients were divided into high (*n* = 91) and low (*n* = 100) groups in the training set, high (*n* = 105) and low (*n* = 91) groups in the testing set, and high (*n* = 196) and low (*n* = 191) groups in the whole set to assess the DSS. The outcomes of patients who had a high score were worse than those with a low score in the training, testing, and whole sets (all *p* < 0.001) (Figure 3G–I). A ROC analysis was conducted on the training (Figure 3J), testing (Figure 3K), and whole set (Figure 3L), and the 1-year, 3-year, and 5-year AUC were 0.702, 0.724, and 0.733; 0.639, 0.602, and 0.653; and 0.672, 0.663, and 0.692, respectively. Finally, we showed that patients with a high score (*n* = 104) had a shorter PFI than patients with a low score (*n* = 105) in the training set, patients with a lower score (*n* = 94) had a significantly longer DFI than patients with a higher score (*n* = 112) in the testing set, and patients with a higher score (*n* = 216) had a significantly lower PFI than patients with a lower score (*n* = 199) (all *p* < 0.001) (Figure 3M–O) in the whole set. In addition, we calculated the 1-year, 3-year, and 5-year AUC of DFI in the training set (0.722, 0.731, and 0.749) (Figure 3P), testing set (0.664, 0.597, and 0.645) (Figure 3Q), and whole set (0.697, 0.666, and 0.695) (Figure 3R).

Next, we analyzed the correlations between protein risk scores and clinical indicators associated with ovarian cancer. By excluding samples with missing clinical information or a small number of cases (Grade 1 n = 6 and Grade 4 n = 1), 409 OS samples were analyzed. In this study, the results were stratified into the following subsets: age ≤ 65 subset (n = 274), age > 65 subset (n = 129), Grade 2 subset (n = 53), and Grade 3 subset (n = 350). Based on the protein risk scores, all subset patients were classifiable into low-risk and high-risk groups according to the optimal cutoff score. A Kaplan–Meier analysis showed that high-risk groups were associated with a shorter OS rate for all subsets of patients (Figure 4A–D). We generated Kaplan–Meier survival curves for DFI and discovered that there were no statistically significant differences between the low-risk and high-risk groups of patients older than 65 years (*p* = 0.243, Appendix A). However, the DFI difference between the two groups was statistically significant for patients under 65 years of age (*p* < 0.001, Appendix A). In addition, we found a statistically significant difference between Grade 2 and 3 patients (*p* < 0.001, Appendix A). Patients were stratified according to their age (<65 years and ≥65 years) and Grade (Grades 2 and 3) of DSS and PFI. Additionally, we conducted Kaplan–Meier analyses that found that higher risk scores were associated with poorer DSS and PFI outcomes (Appendix A). It was determined based on a correlation analysis that high-risk scores were associated with high-grade tumors in OS (*p* = 0.0047, Figure 4E). Likewise, we demonstrated that high-grade tumors had higher risk scores in DFI (*p* = 0.0084, Appendix A), DSS (*p* = 0.0053, Appendix A), and PFI (*p* = 0.0049, Appendix A).

### 3.3. Clinical Value of Prognostic-Related Proteins

Using the median expression score of the nine proteins, patients were subdivided into high and low expression. Using Kaplan–Meier survival analysis, we revealed that the data identified nine protein signatures that were significantly associated with OS (AKT_pS473, *p* < 0.001; AR, *p* = 0.007; BETACATENIN, *p* = 0.018; ERCC5, *p* = 0.014; FOXO3A, *p* = 0.004; IGFRb, *p* = 0.010; P38MAPK, *p* = 0.007; RAB11, *p* = 0.001; and Sox2, *p* = 0.010) (Figure 5). Our next step was to investigate the nine prognostic-related proteins related to DFI (Appendix A), DSS (Appendix A), and PFI (Appendix A) to determine whether they revealed a significant correlation with survival outcomes. High AKT_pS473 and ERCC5 expression were strongly linked to poor clinical outcomes, and high AR, BETACATENIN, FOXO3A, IGFRb, P38MAPK, RAB11, and Sox2 expression were significantly associated with a favorable outcome. By utilizing protein co-expression analysis technology, we determined the co-expression of the nine signature proteins. A total of 32 proteins were identified as significantly co-expressed (correlation coefficients > 0.4 and *p* < 0.001) (Figure 6A). According to Figure 6B, BETACATENIN had the strongest positive correlation with AR but the poorest correlation with RAB11 among the nine proteins.

### 3.4. Construction of Nomogram Model

A univariate and multivariate Cox regression analysis was then conducted to investigate whether the signature affected the outcomes independently of other factors. Based on a univariate Cox analysis, both the risk score (HR 1.336, 95% CI 1.230–1.451, *p* < 0.001) and age (HR 1.031, 95% CI 1.019–1.044, *p* < 0.001) were significant. Additionally, a multivariate Cox analysis of the risk score and age revealed that both were independently poor prognostic factors (HR 1.318, 95% CI 1.214–1.432, *p* < 0.001 and HR 1.030, 95% CI 1.017–1.042, *p* < 0.001, respectively) (Figure 7A–B). By leveraging the nomogram as a prognostic tool, we designed one to develop a clinically relevant tool to help clinicians estimate the chances of 1-year, 3-year, and 5-year OS for ovarian cancer patients (Figure 7C). Based on the calibration plot, we could verify that the predicted probability was well matched to the observed probability from the nomogram for 1-year, 3-year, and 5-year OS, demonstrating good prediction prognosis at these points in time (Figure 7D).

### 3.5. Assessment of Prognosis-Related Protein Subgroups

We performed consensus clustering analysis to reveal the significance of a protein signature in ovarian cancer and identify subgroups on the basis of prognostic-related protein expression. CDF and the K-means algorithm are presented as benchmark strategies in Figure 8A–C, and three prognostic-related protein subgroups were identified. Subsequently, we conducted Kaplan–Meier survival analyses on the three clusters and identified significant differences in OS (*p* = 0.022) (Figure 8D). As shown in the plot, cluster 2 had the worst survival prognosis, whereas cluster 3 had the best. For the DFI analysis, there was no statistically significant difference between the three groups (*p* = 0.187) (Figure 8E). Nevertheless, there was a significant difference between the groups in terms of DSS (*p* = 0.014) (Figure 8F) and PFI (*p* = 0.023) (Figure 8G). It is also worth noting that cluster 2 had the worst survival prognosis, whereas cluster 3 had the best among the three clusters.

### 3.6. Analysis of Prognosis-Related Protein Function, Tumor-Infiltrating Immune Cells, and Immune Checkpoints

As part of the functional enrichment analysis, we utilized GSEA to analyze the KEGG pathways using the transcriptome data of the matched ovarian cancer individuals from TCGA database. An analysis of pathways for genes in the prognostic-related protein high-risk score group revealed that the chemokine signaling pathway, cytokine–cytokine receptor interaction, hematopoietic cell lineage, natural killer cell–mediated cytotoxicity, and Toll–like receptor signaling pathway were the five most enriched pathways (Figure 9A). It was also noteworthy that basal cell carcinoma, the Hedgehog signaling pathway, melanogenesis, taste transduction, and the WNT signaling pathway were the five most enriched pathways in the prognostic-related protein low-risk score group (Figure 9B). Significant differences were also found between the high-risk and low-risk groups in terms of the scores of activated B cells, activated CD4 T cells, activated CD8 T cells, activated dendritic cells, CD56dim natural killer cells, eosinophil, gamma delta T cells, immature B cells, immature dendritic cells, MDSCs, macrophages, mast cells, monocytes, natural killer T cells, natural killer cells, neutrophils, plasmacytoid dendritic cells, regulatory T cells, T follicular helper cells, type 1 T helper cells, type 2 T helper cells, effector memory CD4 T cells, memory B cells, central memory CD4 T cells, central memory CD8 T cells, and effector memory CD8 T cells (*p* < 0.05, Figure 9C). Similarly, different immune cell types were significantly different across prognosis-related protein subgroups (*p* < 0.05, Figure 9D). There were abundantly clear differences in the immune checkpoints between the high-risk and low-risk prognostic-related protein groups (*p* < 0.05, Figure 9E). In addition, there was a differential expression of the immune checkpoint genes in the prognostic-related protein C1, C2, and C3 subgroups (*p* < 0.05, Figure 9F). The five most enriched pathways were allograft rejection, the chemokine signaling pathway, cytokine–cytokine receptor interaction, hematopoietic cell lineage, and the intestinal immune network for IgA production in the C1 subgroup; the chemokine signaling pathway, cytokine–cytokine receptor interaction, ECM receptor interaction, focal adhesion, and hematopoietic cell lineage in the C2 subgroup; and basal cell carcinoma, drug metabolism cytochrome P450, the Hedgehog signaling pathway, taste transduction, and WNT signaling pathway in the C3 subgroup (Figure 9G–I).

### 3.7. An Analysis of Proteins and Protein-Coding Genes Using Online Databases

The HPA database revealed that BETACATENIN, ERCC5, and RAB11 had strong staining, AKT_pS473 and P38MAPK had medium staining, FOXO3A had weak staining, and AR and SOX2 (encode Sox2 protein) were not detected in ovarian cancer tissues (Figure 10). We then utilized Sangerbox to determine the protein-coding gene expression, and *AKT1* (encode AKT_pS473 protein), *FCGR1BP* (encode IGFRB protein), *MAPK13* (encode p13MAPK protein), and *RAB11A* (encode RAB11 protein) showed high expression, while *AR* (encode AR protein), *CTNNB1* (encode BETACATENIN protein), *ERCC5* (encode ERCC5 protein), and *FOXO3* (encode FOXO3A protein) had lower expression (Figure 11A). In a similar fashion, the protein-coding genes whose expression was mutated to varying degrees are shown in Figure 11B. According to the TIMER database and as shown in Figure 12, the expressions of the protein-coding genes were significantly correlated (*AR* and *CTNNB1*, R = 0.224; *AR* and *ERCC5*, R = 0.227; *CTNNB1* and *AKT1*, R = 0.202; *CTNNB1* and *ERCC5*, R = 0.277; *FOXO3* and *AR*, R = 0.277; *FOXO3* and *CTNNB1*, R = 0.377; *FOXO3* and *ERCC5*, R = 0.461; *MAPK13* and *AKT1*, R = 0.216; *MAPK13* and *RAB11A*, R = 0.300; *RAB11A* and *AKT1*, R = 0.216; all *p* < 0.001). Considering the marked differences between the subsets in the cell profiles, we selected the EMTAB8107 and GSE154600 single-cell datasets of ovarian cancer to investigate each subset and were able to identify differentially expressed protein-coding genes, finding that these genes were highly expressed using the TISCH database (Figure 13). Our next objective was to decipher the functional states of cancer cells at the single-cell level, and *CTNNB1* had a significant positive relationship with tumor angiogenesis (R = 0.36, *p* < 0.01, Figure 14A); *CTNNB1* (R = −0.35, *p* < 0.01, Figure 14B), *FOXO3* (R = −0.38, *p* < 0.001, Figure 14C), *MAPK13* (R = −0.48, *p* < 0.01, Figure 14D), and *RAB11A* (R = −0.33, *p* < 0.0, Figure 14E) were shown to have a significant negative relationship with tumor invasion; and *RAB11A* and quiescence were positively correlated (R = 0.31, *p* = 0.01, Figure 14F).

## 4. Discussion

A model that estimates the likelihood of ovarian cancer with the best association with outcomes was developed with nine proteins (P38MAPK, RAB11, FOXO3A, AR, BETACATENIN, Sox2, IGFRb, AKT_pS473, and ERCC5) using a LASSO–Cox algorithm. We found that prognosis-related protein expression was widely variable in ovarian cancer and normal tissues. There was a high-risk group and a low-risk group for ovarian cancer, according to the protein risk scores. A robust difference in OS, DFI, DSS, and PFI curves was observed between the high- and low-score groups in the training, testing, and whole sets. Moreover, we demonstrated close correlations between the protein risk scores and clinical indicators associated with ovarian cancer. Using the median expression scores of the nine proteins, patients were divided into high or low expression. Similarly, the nine protein expression signatures were significantly associated with OS, DFI, DSS, and PFI. Accordingly, for the consensus molecular subtyping of prognostic-related proteins subtypes, they were classified as C1, C2, and C3 clusters, among which C3 showed evidence of OS, DSS, and PFI dominance, followed by cluster C1, whereas cluster C2 suffered the worst clinical outcomes in all ovarian cancer patients.

Through numerous signaling pathways and networks, proteins can promote cell proliferation and migration, contributing to tumorigenesis and cancer progression [25,26]. A cell’s proteome is a collection of functional molecules, and the central dogma dictates how genetic information flows between the genome, epigenome, and proteome. To gain a better understanding of the disease mechanism and develop effective treatments, cell type and developmental stage-specific roles of the genetic regulation of proteins must be investigated [27,28]. Assigning optimal treatments in stratification based on protein signatures in clinical trials and the immune system has the potential to provide insights into the influence of specific proteomic and genomic features. Extensive research has been conducted on proteomic signatures in various malignancies. However, relatively fewer studies have been conducted in patients with ovarian cancer.

Generally, the tumor microenvironment has a strong association with cancer proteomics, confirming its role as a sensitive modulator and target for immunotherapy, and this sheds light on the clinical translation of ovarian cancer [29]. Moreover, the interaction between immune effector cells and malignant cells in the ovarian tumor microenvironment is crucial for cancer survival and metastasis [30,31]. Therefore, we also demonstrated significant differences in immune checkpoint expression and tumor-infiltrating immune cells between high-risk and low-risk prognostic-related protein groups. In patients with specific prognostic-related protein subtypes of ovarian cancer, we observed significant differences in the expression of immune checkpoint genes and tumor-infiltrating immune cells between high- and low-risk groups. It became increasingly apparent that the TME played a significant role in the immune function and could influence and be influenced by adjacent and embedded immune cells among the different clusters. Specifically, the effectiveness of cancer immunotherapy has been significantly enhanced with the development of immune checkpoint modulators and the use of immune effector cells [32,33,34,35]. In this study, we demonstrated that among patients with high-risk and low-risk prognostic-related protein groups, as well as different prognostic-related protein clusters, there were distinguishable signaling pathways enriched for those involved in immune-related and other signaling pathways, consistent with previous studies [30,36,37,38,39].

There was a high likelihood that the signatures of nine proteins and co-expressed proteins may strongly overlap. Furthermore, we found that protein-coding genes were widely expressed in both whole tissues and single-cell subsets, had varying degrees of mutations, and were significantly correlated with each other. Moreover, our findings confirm that genes played a role in the functional states of cancer cells at the single-cell level. Among the nine protein-coding genes, *BETACATENIN* showed the strongest positive correlation with *AR* but the weakest correlation with *RAB11*. Therefore, our nomogram of protein signatures and clinical data could be a valuable prognostic tool to help clinicians assess patient survival. Consistent with our findings, a previous study found that protein signatures were related to prognosis. A study proposed an autophagy-mediated protein degradation mechanism to arrest protein maturation. It is suggested that the combination of the PIKfyve inhibitor and p38MAPK is best suited for colorectal adenocarcinoma [40]. Moreover, *RAB11* activity has been shown to promote cancer cell invasion and metastasis through distinct mechanisms and predict poor survival in various cancers, and there is evidence to suggest that surface proteomes of cancer cells are regulated by Rab11 [41,42]. Interestingly, the current study identified that *FOXO3a* played an important role in multiple cell signaling pathways, including WNT/β-catenin signaling. Moreover, cancer cells are induced or maintained by transcription factors that are directly responsible for the pathogenesis of diverse tumors [43,44]. Furthermore, this study discovered evidence linking protein signatures with immune cells in the tumor–immune environment and immune-associated pathways. We also found that *AR* could bind directly to the *IL12A* promoter region to repress transcription, thereby suppressing the cytotoxicity of IL-12-activated NK cells in HCC [45]. As has been highlighted by scholars, ovarian cancer cell proliferation can also be mediated by the androgen/AR-independent activation of PI3K/AKT [46].

The transcription factor *Sox2* played an important role in maintaining embryogenesis, which was required for effective pluripotency and reprogramming. There is considerable evidence to suggest that *Sox2* is critical for maintaining ovarian cancer stem cell pluripotency as well as determining stem cell fate, and targeting *Sox2* may be therapeutically beneficial [47,48]. However, the inhibition of AKT_pS473-activated *FOXO3a*, in turn, exhibited potent antiproliferative, apoptotic, and tumor-suppressing properties. At least some of the observed phenomena were attributed to restricting the expression of *AR* and 5α-reductase [49]. According to other findings, *ERCC5* was a novel prognostic indicator for predicting ovarian cancer survival and a potential target in platinum chemotherapy [50]. In particular, it has been found that polymorphisms in the *ERCC5* gene are associated with cancer risk [51]. These findings demonstrate that prognostic-related protein signatures may play an important role in ovarian cancer onset and progression, suggesting that these signatures may be used as prognostic biomarkers. It is, therefore, necessary to investigate the cellular and molecular functions of these prognostic-related protein signatures in ovarian cancer in the future.

There are some limitations to this study as it is a bioinformatic study based on data collected through public databases. First, although we have obtained more reliable findings through the integrated analyses of transcriptomic and proteomic data, and we acknowledge that the further integration of multi-omics data (methylation, mi-RNA, lncRNA, and metabolomics) could provide a more comprehensive understanding of the complex molecular basis of ovarian cancer. Second, we acknowledge that a limitation of this study is that clinical information on participants was extracted from collaborative initiatives such as TCGA rather than gathering our own data to explore the complex layered omics basis of ovarian cancer. Third, while it is unrealistic to perform a prospective study to validate the study results, alternatively, we could select an interaction profile for pre-clinical studies. With the proposal of precision medicine, treatment targeting specific genes and targets has become a certain development trend. The treatment of the end product protein of gene translation will also provide a new direction for the treatment of ovarian cancer. Testing the expression of prognostic-related proteins in ovarian cancer patients can help predict their prognosis and timely follow-up. In case of disease recurrence or metastasis, early intervention can improve the prognosis and increase the survival rate.

## 5. Conclusions

To summarize, we have reported and validated a survivability prediction model for ovarian cancer prepared on the basis of prognostic-related protein signatures. We investigated the relationship between the signatures, tumor-infiltrating immune cells, and immune checkpoints, and conclude that the heterogeneity and complexity of the TME are caused by the regulatory patterns of different protein signatures. This lays the groundwork for understanding immunomodulation in ovarian cancer. Furthermore, we used single-cell RNA and bulk RNA sequencing to find that protein-coding genes are differentially expressed. Additionally, the genes have relationships with each other and tumor functional states.

## Figures and Tables

**Figure 1 biomolecules-13-00685-f001:**
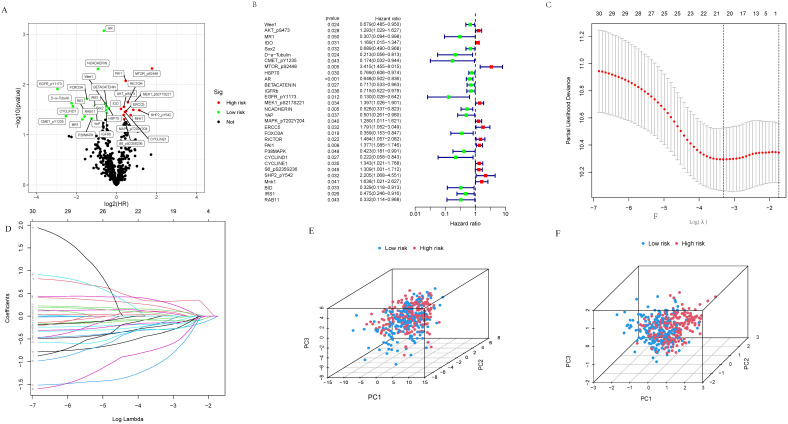
Identification of proteins with prognostic relevance in ovarian cancer. (**A**) A volcano plot was created for the prognosis of differential proteins via univariate Cox regression analyses. The green points show the 18 low-risk proteins, and the red points illustrate the 12 high-risk proteins. (**B**) A forest plot for each prognostic factor was created. The green show the low-risk proteins, and the red illustrate the high-risk proteins. (**C**) The deviance profiles of the LASSO regressions. (**D**) The coefficient profiles of the LASSO regressions. Diverse hues of lines signif distinct gene. (**E**) Analysis of all proteins using PCA. The blue show the low-risk proteins, and the red illustrate the high-risk proteins. (**F**) Analysis of prognostic proteins using PCA. The blue show the low-risk proteins, and the red illustrate the high-risk proteins.

**Figure 2 biomolecules-13-00685-f002:**
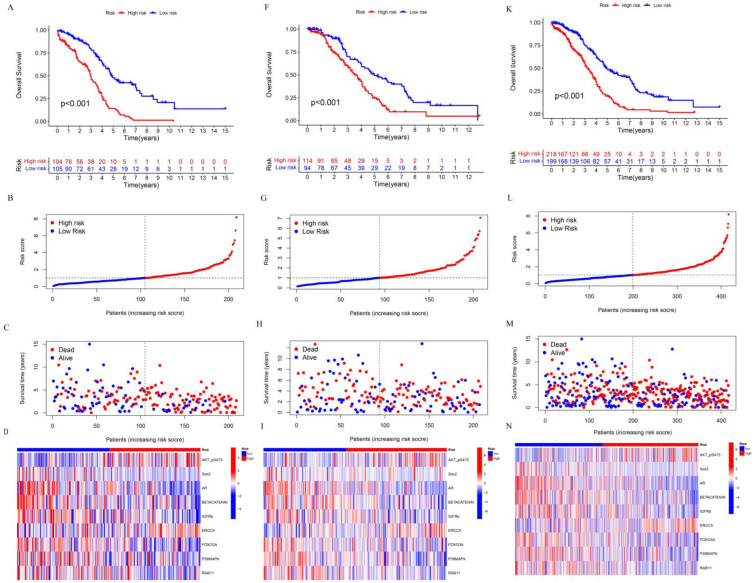
Assessment of prognosis-related protein model for overall survival (OS). (**A**) Kaplan–Meier survival curve of OS between high- and low-score groups in the training, testing (**F**), and whole (**K**) sets. (**B**) The distribution of the risk score of prognosis-related proteins in the training, testing (**G**), and whole (**L**) sets. (**C**) The distribution of survival status of prognosis-related proteins in the training, testing (**H**), and whole (**M**) sets. (**D**) The expression level of prognosis-related proteins in the training, testing (**I**), and whole (**N**) sets. (**E**) ROC curve for 1-year, 3-year, and 5-year survival rates in the training, testing (**J**), and whole (**O**) sets.

**Figure 3 biomolecules-13-00685-f003:**
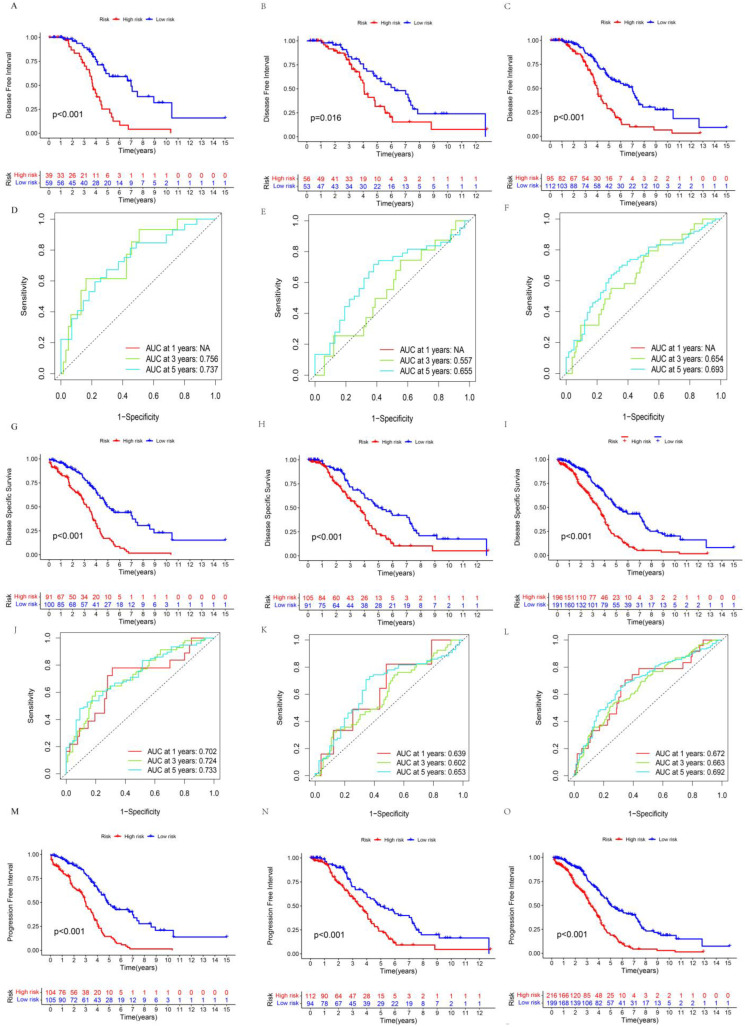
Assessment of prognosis-related protein model for disease-free interval (DFI), disease-specific survival (DSS), and progression-free interval (PFI). (**A**) Kaplan–Meier survival curve for DFI between high- and low-score groups in the training, testing (**B**), and whole (**C**) sets. (**D**) ROC curve for 1-year, 3-year, and 5-year DFI in the training, testing (**E**), and whole (**F**) sets. (**G**) Kaplan–Meier survival curve for DSS between high- and low-score groups in the training, testing (**H**), and whole (**I**) sets. (**J**) ROC curve for 1-year, 3-year, and 5-year DSS in the training, testing (**K**), and whole (**L**) sets. (**M**) Kaplan–Meier survival curve for PFI between high- and low-score groups in the training, testing (**N**), and whole (**O**) sets. (**P**) ROC curve for 1-year, 3-year, and 5-year PFI in the training, testing (**Q**), and whole (**R**) sets.

**Figure 4 biomolecules-13-00685-f004:**
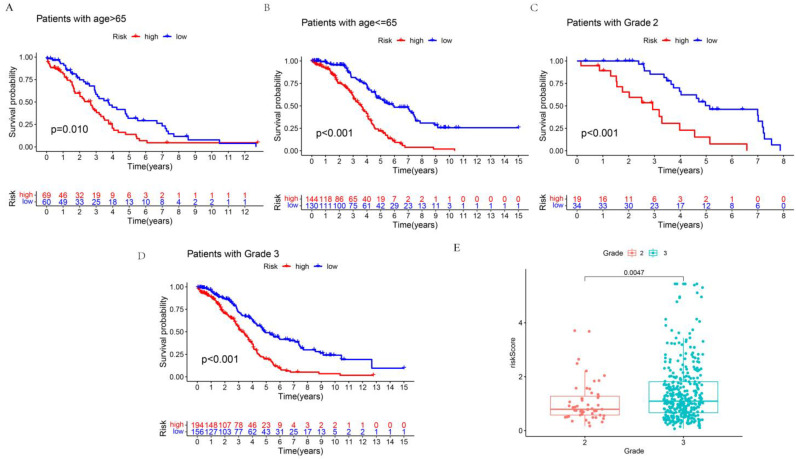
Correlations between protein risk score and clinical indicators. (**A**) Kaplan–Meier survival curve for overall survival (OS) in patients age >65 years. (**B**) Kaplan–Meier survival curve for OS in patients age ≤65 years. (**C**) Kaplan–Meier survival curve for OS in patients with Grade 2. (**D**) Kaplan–Meier survival curve for OS in Grade 3 patients. (**E**) Scatter plot showing the correlation between risk score and grade in OS.

**Figure 5 biomolecules-13-00685-f005:**
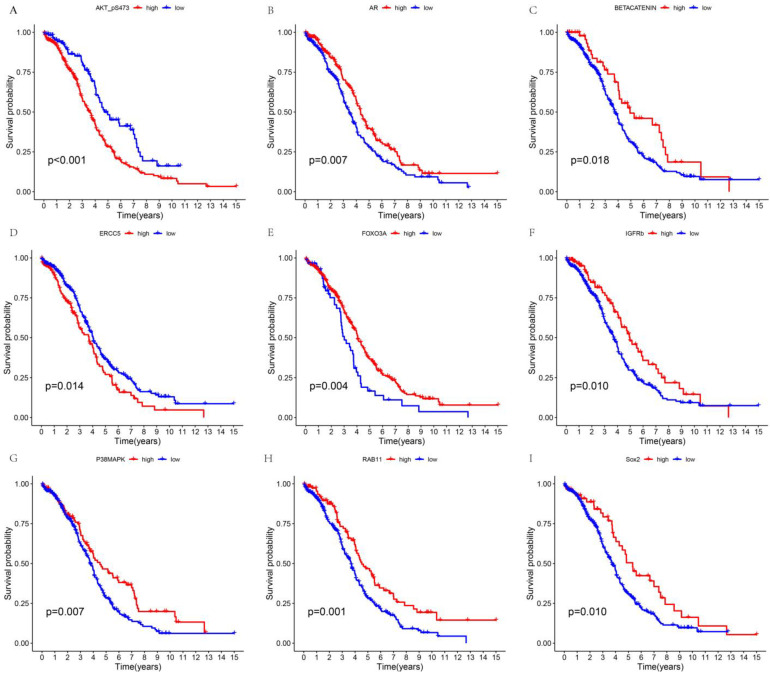
Kaplan–Meier survival curve of overall survival (OS) between high and low expression of the nine protein signatures. (**A**) Kaplan–Meier survival curve of OS between high- and low-expression of AKT_pS473.(**B**) Kaplan–Meier survival curve of OS between high- and low-expression of AR. (**C**) Kaplan–Meier survival curve of OS between high- and low-expression of BETACATENIN. (**D**) Kaplan–Meier survival curve of OS between high- and low-expression of ERCC5. (**E**) Kaplan–Meier survival curve of OS between high- and low-expression of FOXO3A. (**F**) Kaplan–Meier survival curve of OS between high- and low-expression of IGFRb. (**G**) Kaplan–Meier survival curve of OS between high- and low-expression of P38MAPK. (**H**) Kaplan–Meier survival curve of OS between high- and low-expression of RAB11. (**I**) Kaplan–Meier survival curve of OS between high- and low-expression of Sox2.

**Figure 6 biomolecules-13-00685-f006:**
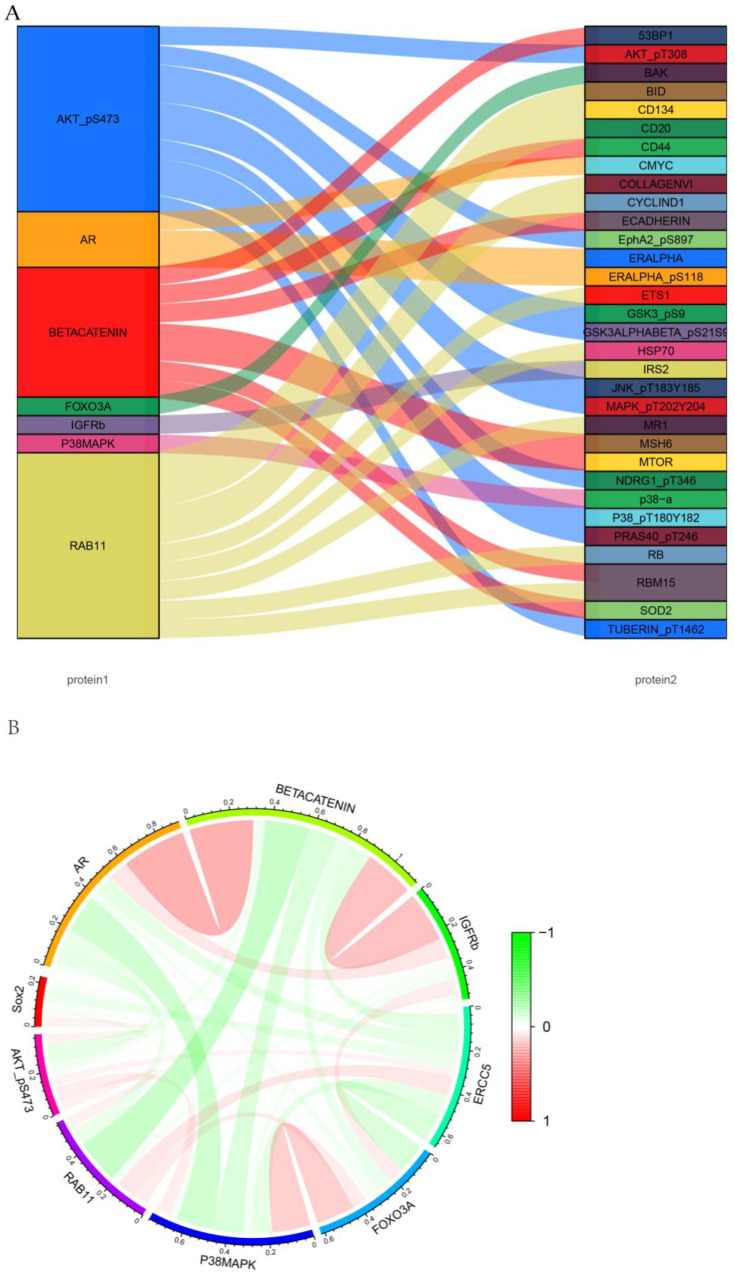
Analysis of co-expressed proteins and relationships between proteins. (**A**) Sankey diagram of the co-expression of the nine signature proteins. (**B**) Circle plot showing the relationships between the nine signature proteins.

**Figure 7 biomolecules-13-00685-f007:**
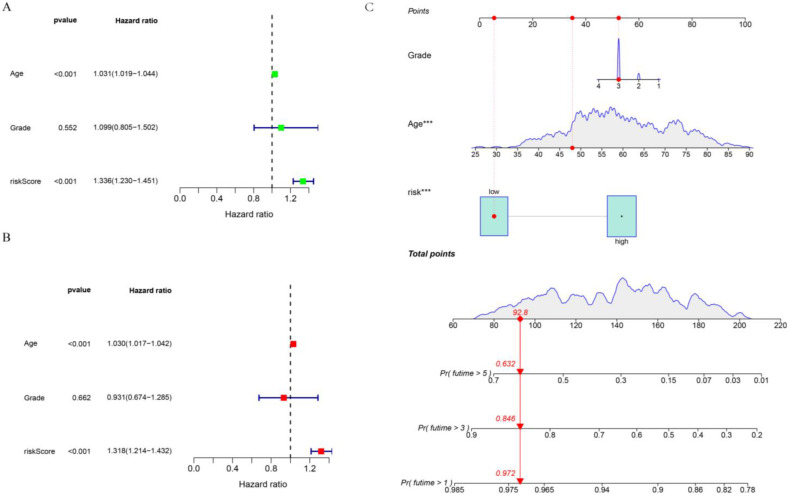
Construction of nomogram model. (**A**) Univariate Cox regression analyses were performed using the risk score and other clinicopathological variables. The green color denotes the point estimates of hazard ratio (HR) for each variable. (**B**) Multivariate Cox regression analyses were performed using the risk score and other clinicopathological variables. The red color denotes the point estimates of hazard ratio (HR) for each variable. (**C**) The nomogram model was constructed based on the risk score and other clinicopathological variables. The upper part of the diagram uses red color to indicate the score values associated with the variable, while the lower part shows red color to represent the survival rates for 1, 3, and 5 years that correspond to a total score of 92.8. (**D**) The nomogram’s accuracy was assessed using calibration curves. The green color represents the calibration curve of 1 year. The blue color represents the calibration curve of 3 years. The red color represents the calibration curve of 5 years (*** *p* < 0.001).

**Figure 8 biomolecules-13-00685-f008:**
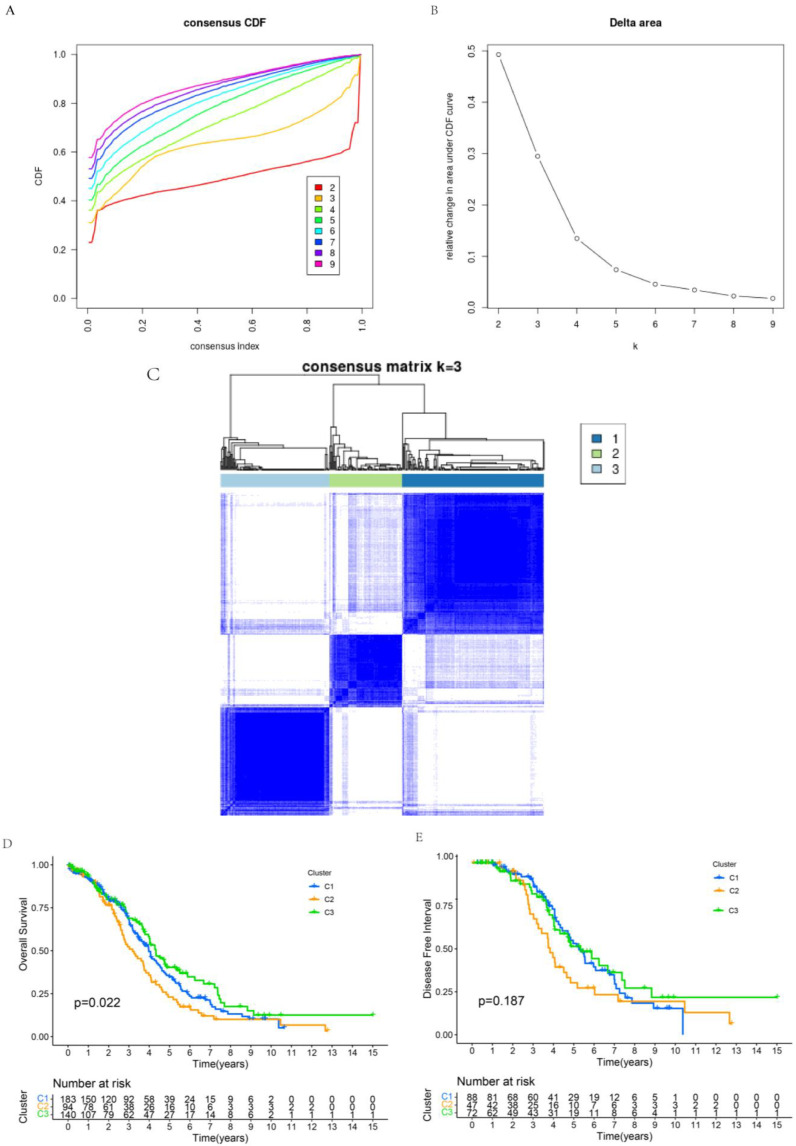
Identification of prognosis-related protein subgroups. (**A**) The CDF curve. (**B**) The K-means algorithm. (**C**) The consensus matrices of three prognostic-related protein subgroups. (**D**) Kaplan–Meier survival curve of prognostic-related protein subgroup overall survival (OS). (**E**) Kaplan–Meier survival curve of prognostic-related protein subgroup disease-free interval (DFI). (**F**) Kaplan-Meier survival curve of prognostic-related protein subgroup disease-specific survival (DSS). (**G**) Kaplan–Meier survival curve of prognostic-related protein subgroup progression-free interval (PFI).

**Figure 9 biomolecules-13-00685-f009:**
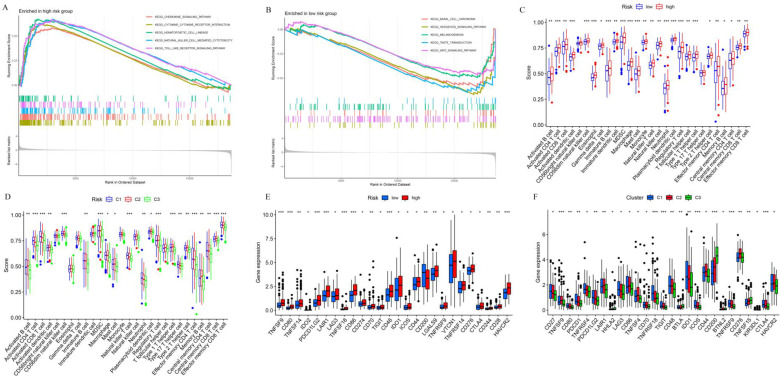
Assessment of prognosis-related protein function, tumor-infiltrating immune cells, and immune checkpoints. (**A**) GSEA analysis of the five pathways in the high-risk score group. (**B**) GSEA analysis of the five pathways in the low-risk score group. (**C**) Tumor-infiltrating immune cell analysis between the high-risk and low-risk groups. (**D**) Tumor-infiltrating immune cell analysis across prognostic-related protein subgroups. (**E**) Immune checkpoint analysis between the high-risk and low-risk groups. (**F**) Immune checkpoint analysis across prognostic-related protein subgroups. (**G**) GSEA analysis of the five pathways in C1. (**H**) GSEA analysis of the five pathways in C2. (**I**) GSEA analysis of the five pathways in C3 (* *p* < 0.05, ** *p* < 0.01, *** *p* < 0.001).

**Figure 10 biomolecules-13-00685-f010:**
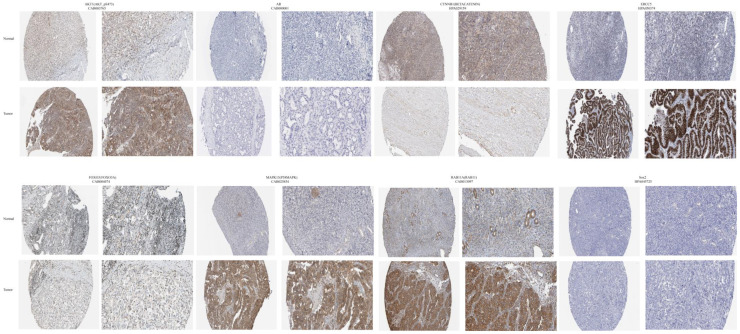
Expression of prognosis-related protein in ovarian cancer and normal tissues in HPA database.

**Figure 11 biomolecules-13-00685-f011:**
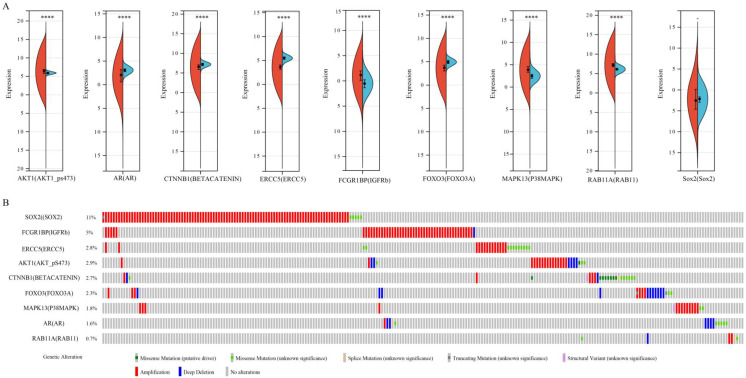
Analysis of protein-coding genes. (**A**) Expression of protein-coding genes between ovarian cancer and normal tissues in Sangerbox. (**B**) The mutation of protein-coding genes in cBioPortal database (**** *p* < 0.0001).

**Figure 12 biomolecules-13-00685-f012:**
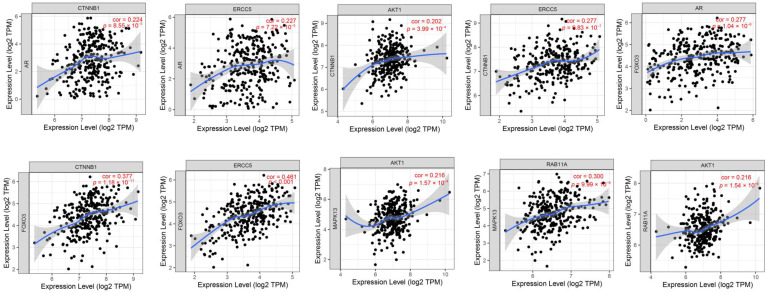
The correlation of the protein-coding genes in the TIMER database.

**Figure 13 biomolecules-13-00685-f013:**
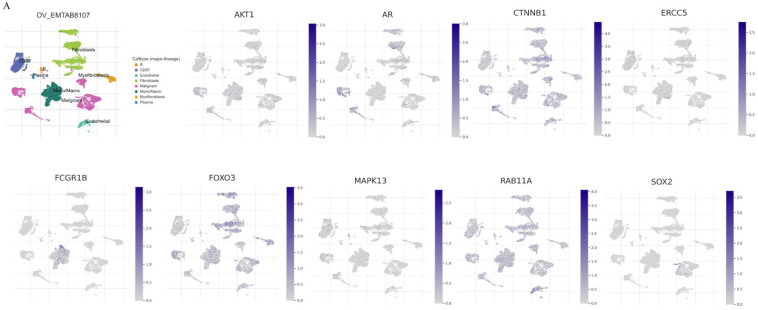
Expression of protein-coding genes in single-cell datasets of ovarian cancer. (**A**) Protein-coding gene expression in EMTAB8107. (**B**) Protein-coding gene expression in GSE154600.

**Figure 14 biomolecules-13-00685-f014:**
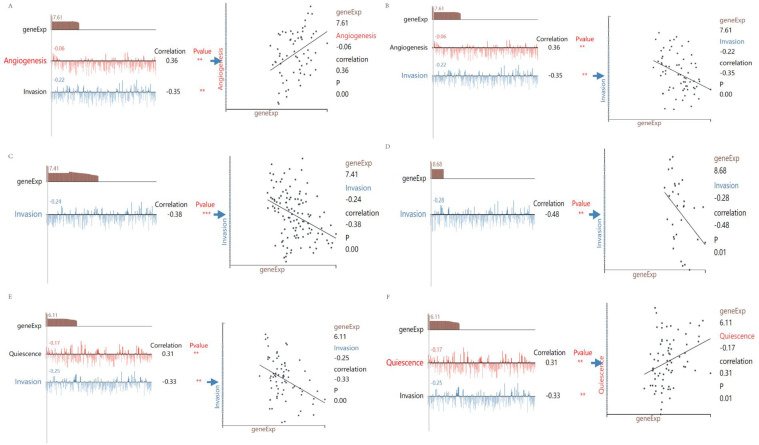
The correlation of the functional states of cancer cells and protein-coding genes at the single-cell level in the CancerSEA database. (**A**) The relationship of *CTNNB1* with tumor angiogenesis. (**B**–**E**) The relationship of *CTNNB1*, *FOXO3*, *MAPK13*, and *RAB11A* with tumor invasion. (**F**) The relationship of *RAB11A* with tumor quiescence (** *p* < 0.01, *** *p* < 0.001).

## Data Availability

Proteome profiling and clinical and transcriptome data about ovarian cancer were downloaded from TCGA Database (https://portal.gdc.cancer.gov/). Please contact the corresponding author if you want to access the analysis codes.

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
