# Peer review of "Screening and Identification of a Prognostic Model of Ovarian Cancer by Combination of Transcriptomic and Proteomic Data"

_biomolecules, 2023, doi:10.3390/biom13040685_

Round 1

Reviewer 1 Report

Excellent piece of bioinformatic work. The main issue with all these computational experiments remains the biologic implementation. This effort is not new by the way. I suggest the authors make a clear strong statement at the beginning of their discussion section about the main findings of this study. To me, this should be the construction of a prognostic model incorporating inter-relationship between transcriptomics and proteomics features, which can lead to an enhanced global view to explaining the ovarian cancer phenotype. Actually, can they tell from these experiments whether integrating -omics data is better than non-integrating _not exploring the inter-relationship between -omics data)? And, if this is the case, they then need to mention that further integration of -multi omics data (methylation, mi-RNA, metabolomics) could potentially improve our understanding of cancer or even extracting new biological knowledge. (this needs to be mentioned in the limitations of the study).

I remain uneasy with the extraction of clinical information from collaborative initiatives such as the TCGA, instead of own data (real world data) to explore the complex layered -omics basis of cancer (this is yet a limitation). Also, I believe that it is unrealistic to perform a prospective study to validate the study results but alternatively, to select interaction profile for pre-clinical studies.

Minor points

Why LASSO-based cox regression?

You mean Stage 3-4 than Grade 3-4? Unless you call grade 4 undifferentiated ovarian neoplasms (rather rare). The routine is we use grading in ovarian cancer as follows: grade 1(low grade) and grade 2-3(high grade).

I feel English needs moderate to extensive editing.

Author Response

Point 1: Excellent piece of bioinformatic work. The main issue with all these computational experiments remains the biologic implementation. This effort is not new by the way. I suggest the authors make a clear strong statement at the beginning of their discussion section about the main findings of this study. To me, this should be the construction of a prognostic model incorporating inter-relationship between transcriptomics and proteomics features, which can lead to an enhanced global view to explaining the ovarian cancer phenotype. Actually, can they tell from these experiments whether integrating -omics data is better than non-integrating _not exploring the inter-relationship between -omics data)? And, if this is the case, they then need to mention that further integration of -multi omics data (methylation, mi-RNA, metabolomics) could potentially improve our understanding of cancer or even extracting new biological knowledge. (this needs to be mentioned in the limitations of the study). I remain uneasy with the extraction of clinical information from collaborative initiatives such as the TCGA, instead of own data (real world data) to explore the complex layered -omics basis of cancer (this is yet a limitation). Also, I believe that it is unrealistic to perform a prospective study to validate the study results but alternatively, to select interaction profile for pre-clinical studies.

Response 1: Thank you for your advice first. This proposal will make our article more comprehensive. We are very grateful to the reviewer for reviewing the paper so carefully. We have made a clear strong statement at the beginning of the discussion section according to your recommendations. Our findings apply to construct a clinically useful model that incorporates transcriptomic and proteomic features to predict the prognostic of ovarian cancer and thereby lead to an enhanced global view to explaining the ovarian cancer phenotype. First, we uncover prognostic-related proteins and develop a new protein prognostic signature for patients with ovarian cancer to predict their prognosis. Patients were brought together in subgroups using a consensus clustering analysis of prognostic-related proteins. And, final resulting prognosis factors consisted of seven protective factors (P38MAPK, RAB11, FOXO3A, AR, BETACATENIN, Sox2, and IGFRb) and two risk factors (AKT_pS473 and ERCC5), which can be used to construct a prognosis-related protein model. A significant difference in OS, DFI, DSS, and PFI curves were found in the training, testing, and whole sets when analyzing the protein-based risk score Second, we also illustrated a wide range of functions, immune checkpoints, and tumor-infiltrating immune cells in prognosis-related protein signatures and protein-coding genes signatures. Additionally, proteins and protein-coding genes were analyzed using online databases (HPA, Sangerbox, TIMER, cBioPortal, TISCH, and CancerSEA). You point to a number of limitations of this paper, we have made a statement in our article according to your kindly suggestions. We have added some information in on the limitations of the study: As a bioinformatics study based on data collected through public databases, there are some limitations to this study. First, we have obtained more reliable findings through integrated analyses of transcriptomic and proteomic data, and we do not mention the further integration of multi-omics data (methylation, mi-RNA, LncRNA, metabolomics). Second, a shortcoming is the extraction of clinical information on participants from collaborative initiatives such as TCGA rather than gathering our own data to explore the complex layered omics basis of ovarian cancer. Third, it is unrealistic to perform a prospective study to validate the study results, but alternatively, we could select an interaction profile for pre-clinical studies. Thank you for pointing out the limitations of the article, so as to add a certain weight to the reliability of the article.

Minor points

Point 1: Why LASSO-based cox regression?

Response 1: Thank you for your thoughtful suggestions. Significance of survival variables is analyzed by the stepwise Cox proportional hazards model. Next, multiple proteins are combined into panels, to evaluate if any particular combination of proteins exhibited further be associated with prognosis and developed new therapeutic targets of ovarian cancer, using Lasso regression analysis. In order to determine the proteins biomarkers of ovarian cancer, we use a method that has previously been published:(1) Wang X, Wu L, Ai L, et al. Construction of an HCC recurrence model based on the investigation of immune-related lncRNAs and related mechanisms. Molecular Therapy - Nucleic Acids. 2021, 26: 1387-1400. (2) Shi Y, Wang J, Huang G, et al. A novel epithelial–mesenchymal transition gene signature for the immune status and prognosis of hepatocellular carcinoma. Hepatology International. 2022,16:906-917. (3) Mo J, Cui Z, Wang Q, et al. Integrated analysis of necroptosis-related lncRNAs for prognosis and immunotherapy of patients with pancreatic adenocarcinoma. Frontiers in Genetics. 2022, 13:940794.

Point 2: You mean Stage 3-4 than Grade 3-4? Unless you call grade 4 undifferentiated ovarian neoplasms (rather rare). The routine is we use grading in ovarian cancer as follows: grade 1(low grade) and grade 2-3(high grade).

Response 2: We are very grateful for your valuable suggestions. Indeed, we have neglected relatively scarce of the Grade 4 undifferentiated ovarian cancer. We carefully check the clinical data of TCGA for each ovarian cancer patients again to clarify the effect of analysis. Only one patient is undifferentiated ovarian neoplasm (Grade 4), six patients are low grade (Grade 4). To assure the reliability of this results, we exclude the small number of cases (Grade 1 n=6, Grade 4 n=1). For a reliable evaluation, we then use diverse grades (Grade 2 and 3) of ovarian cancer patients of TCGA in our analysis.

Point 3: I feel English needs moderate to extensive editing.

Response 3: Thank you for your thoughtful suggestions. We regret there are problems with the English. The paper has been carefully revised by English Editing Department of MDPI (https://www.mdpi.com/authors/english) to improve the grammar and readability. MDPI English Editing points that the text has been checked for correct use of grammar and common technical terms,  and edited to a level suitable for reporting research in a scholarly journal.

Reviewer 2 Report

In the article entitled “Screening and identification a prognostic model of ovarian cancer by combination transcriptomics and proteomics data” the authors describe a prognostic signature composed of 9 proteins. This was constructed from publicly available data (TCGA) using a regression model and then confirmed in additional datasets. While the results are of interest to the community, as currently presented the paper is very hard to follow. There is a mixture between tenses throughout the paper and inappropriate use of words or phrases that clouds the message of the paper. A better description of which data is analyzed in the results section is needed. For example, it is not stated in the first paragraph of the results section (lines 177-191), which data is used or what two groups the authors are examining. Secondly, in this section the authors refer to proteins known to associate with prognosis, however, how these were curated or gathered is not described in the results. This makes the text hard to follow. In addition, figure legends can be improved (for example, some lack description of colors) and the resolution of the figures should be improved.

Throughout the introduction and abstract there are issues with grammar. For example, it should be transcriptomic and proteomic data.

Example of how to correct the first two sentences of the paper:

Women with ovarian cancer become victims of one of the deadliest gynecological malignancies in the world[1-3]. The Global Cancer Statistics 2020 reported 313,959 confirmed cases of ovarian cancer had resulting in 207,252 disease-related deaths [4].

On line 42 it is unclear what cancer profiles are being referred to.

Author Response

Point 1: In the article entitled “Screening and identification a prognostic model of ovarian cancer by combination transcriptomics and proteomics data” the authors describe a prognostic signature composed of 9 proteins. This was constructed from publicly available data (TCGA) using a regression model and then confirmed in additional datasets. While the results are of interest to the community, as currently presented the paper is very hard to follow. There is a mixture between tenses throughout the paper and inappropriate use of words or phrases that clouds the message of the paper. A better description of which data is analyzed in the results section is needed. For example, it is not stated in the first paragraph of the results section (lines 177-191), which data is used or what two groups the authors are examining. Secondly, in this section the authors refer to proteins known to associate with prognosis, however, how these were curated or gathered is not described in the results. This makes the text hard to follow. In addition, figure legends can be improved (for example, some lack description of colors) and the resolution of the figures should be improved.

 Response 1: Thank you for your letter and for the reviewer’ comments concerning our manuscript entitled “Screening and identification a prognostic model of ovarian cancer by combination transcriptomic and proteomic data”. Those comments are all valuable and very helpful for revising and improving our paper, as well as the important guiding significance to our researches. We have studied comments carefully and have made correction which we hope meet with approval. Revised portions are marked in red in the paper. Thanks very much for taking your time to review this manuscript. We really appreciate all your generous comments and suggestions! We regret there are problems with the English and there are issues with grammar and inappropriate with words or phrases. The paper has been carefully revised by English Editing Department of MDPI(https://www.mdpi.com/authors/english) to improve the grammar and readability. MDPI English Editing point that the text has been checked for correct use of grammar and common technical terms, and edited to a level suitable for reporting research in a scholarly journal. The data sources used have been detailed in the results section, so that the readers can have a clearer understanding of the basic information of the data. We have described the proteins data in the Materials and Methods part(2.1.Extraction of data: Proteome profiling data about ovarian cancer were downloaded from TCGA’s database (https://portal.gdc.cancer.gov/) [18]. For the acquisition of the corresponding clinical and transcriptome data of matched ovarian cancer individuals, we used TCGA’s database.) We further characterized the proteins associated with prognosis (see 3. Results: 3.1. Proteins with prognostic relevance in ovarian cancer: A simultaneous analysis of the forest plots of the prognostic factors of HR and the 95% confidence intervals of the risk rates for every single protein is presented in Figure 1B. A total of 30 protein biomarkers were input into the subsequent LASSO regression correlation analysis of ovarian cancer (Figure 1C,D), and 20 proteins with non-zero coefficients were identified. As a final step, we performed a multivariate analysis when including 20 proteins known to be prognosis factors into the analysis. The final resulting prognosis factors consisted of seven protective factors (P38MAPK, RAB11, FOXO3A, AR, BETACATENIN, Sox2, and IGFRb) and two risk factors (AKT_pS473 and ERCC5)). In addition, we have improved the figure legends to make what explains clearer and easier to understand in results section. As valuable recommended by the reviewer, we have improved the the resolution of the figures. Finally, we thank you for your consideration of our revised manuscript.

Point 2:Throughout the introduction and abstract there are issues with grammar. For example, it should be transcriptomic and proteomic data.

Example of how to correct the first two sentences of the paper:

Women with ovarian cancer become victims of one of the deadliest gynecological malignancies in the world[1-3]. The Global Cancer Statistics 2020 reported 313,959 confirmed cases of ovarian cancer had resulting in 207,252 disease-related deaths [4].

On line 42 it is unclear what cancer profiles are being referred to.

Response 2: Thank you for your thoughtful suggestions. We regret there are problems with the English and grammar. The paper has been carefully revised by English Editing Department of MDPI (https://www.mdpi.com/authors/english) to improve the grammar and readability. MDPI English Editing point that the text has been checked for correct use of grammar and common technical terms, and edited to a level suitable for reporting research in a scholarly journal.

Round 2

Reviewer 2 Report

Review for the article by Jiang et al. “Screening and Identification of a Prognostic Model of Ovarian Cancer by Combination Transcriptomic and Proteomic Data.”

The paper by Jiang et al uses publicly available data to make a signature based on protein signature to predict the prognosis of women with ovarian cancer. The results are interesting and seem relevant to the community. However, the message of the paper is often lost in the wording, making the it difficult to recommend the paper for publication.

Here are several suggestions:

1.     Fix the title: “Screening and Identification of a Prognostic Model of Ovarian Cancer by Combination of Transcriptomic and Proteomic Data.”

2.     Use standard gene or protein names or at least reference these once

3.     In the abstract and the paper OS, DFI, DSS and PFI are used without defining the abbreviations.

4.     Ln 56 should be genomes or change the other omics

5.     Ln 108 should be risk not risky

Questions regarding the content:

Ln 51, the Authors state, “In order to improve ovarian cancer patients’ prognosis, it is essential to screen patients at higher risk for accurate prognosis.”

It is unclear to the reviewer what the authors mean by this sentence.

Ln 60: Some scholars have gained a better understanding of proteins’ functions in the proteome of ovarian carcinoma to gain a more comprehensive understanding, provid- ing new avenues for developing more effective diagnostics and therapies [10,11].

This sentence is unclear.

Ln 179 The univariate Cox regression analyses of prognosis for differential proteins showed that they were significantly associated with outcome and could be a powerful indicator for the training set when using the proteome profiling data about ovarian cancer from the TCGA database.

Does this mean that the authors performed univariate Cox regression to identify differential proteins?

Ln 189, it is unclear where the additional 20 proteins came from? What are prognosis factors? Do you mean factors that associate with or predict prognosis?

Ln 205 Is it the construction of a protein signature of prognosis?

Ln 235. This sentence is confusing.

Ln 496 which study are you referring to? The phrase “The present study” makes it sound like the one the authors have submitted for review, but it is not in line with the text.

Ln 340 Prognosis-related proteins are a robust tool implemented in ovarian cancer for patient prognostication. Which proteins are these? Are they the signature discussed here?

How do the protein models compare to transcriptomic based models?

The term the authors use prognostic-related protein risk score is confusing. Perhaps it would be better to say protein-risk score only?

Ln 488 furthermore is spelled wrong

Ln 490 mutation is spelled wrong

In the discussion how this would be implemented in the clinic could be discussed

Author Response

Response to Reviewer Comments

Comments and Suggestions for Authors

Point 1: The paper by Jiang et al uses publicly available data to make a signature based on protein signature to predict the prognosis of women with ovarian cancer. The results are interesting and seem relevant to the community. However, the message of the paper is often lost in the wording, making it difficult to recommend the paper for publication.

Response 1: Thank you for reading the article extensively and providing thoughtful suggestions. We appreciate your affirmation of the article's content, and it is a great motivation for us to receive your approval. We will make every effort to revise the article to meet the acceptance and publication requirements of your journal. Your encouragement and expectations are especially significant for a doctoral candidate who will be graduating in June this year. Regarding your feedback on language and writing, we have invited the language editing company (English Editing Department of MDPI,https://www.mdpi.com/authors/english) recommended by your journal to make the necessary revisions according to your requirements during the first round of review. During the revision process, we have carefully considered and modified the language and expressions in the article and consulted with English language professionals to meet your expectations and those of your esteemed journal. Once again, we appreciate your efforts in reviewing the article.

Here are several suggestions

Point 1: Fix the title: “Screening and Identification of a Prognostic Model of Ovarian Cancer by Combination of Transcriptomic and Proteomic Data.”

Response 1: Thank you for your valuable suggestion on the modification of the article title. We have made the necessary changes according to your recommendation.

Point 2: Use standard gene or protein names or at least reference these once.

Response 2: Thank you for your valuable suggestions on the manuscript. This is indeed something we overlooked. The names of the proteins were defined during sequencing, and we have represented all encoded proteins genes through website (https://www.uniprot.org/), as well as providing a clear explanation of the protein encoding genes in section 3.7 of the result. It is shown below: The HPA database revealed that BETACATENIN, ERCC5, and RAB11 had strong staining, AKT_pS473 and P38MAPK had medium staining, FOXO3A had weak staining, and AR and SOX2 (encode Sox2 protein) were not detected in ovarian cancer tissues (Figure 10). We then utilized Sangerbox to determine the protein-coding gene expression, and AKT1 (encode AKT_pS473 protein), FCGR1BP (encode IGFRB protein), MAPK13 (encode p13MAPK protein), and RAB11A (encode RAB11 protein) showed high expression, While AR (encode AR protein), CTNNB1 (encode BETACATENIN protein), ERCC5 (encode ERCC5 protein), and FOXO3 (encode FOXO3A protein) had lower expression (Figure 11A).

Point 3: In the abstract and the paper OS, DFI, DSS and PFI are used without defining the abbreviations.

Response 3: Thank you for your sincere suggestion. The lack of definition for the abbreviation did cause confusion for readers. Therefore, following your advice, we have made modifications to the abstract, as shown below: A significant difference in overall survival (OS), disease free interval (DFI), disease specific survival (DSS), and progression free interval (PFI) curves were found in the training, testing, and whole sets when analyzing the protein-based risk score (P<0.05).

Point 4: Ln 56 should be genomes or change the other omics.

Response 4: Thank you for your constructive suggestion. In order to avoid unnecessary misunderstanding, we have removed “omics” as per your suggestion.”

Point 5:  Ln 108 should be risk not risky.

Response 5: This writing error you pointed out is much appreciated. The original sentence is: Eventually, we identified protective (hazard ratio, HR > 1 and P<0.05) and risky proteins (HR < 1 and P<0.05). And the modified sentence is: Eventually, we identified proteins with significant associations with prognosis, either as protective (hazard ratio, HR > 1 and P<0.05) or as risk proteins (HR < 1 and P<0.05).

Questions regarding the content

Point 1: Ln 51, the Authors state, “In order to improve ovarian cancer patients’ prognosis, it is essential to screen patients at higher risk for accurate prognosis.”

It is unclear to the reviewer what the authors mean by this sentence.

 Response 1: Thank you for your valuable revision suggestion, reviewer. Upon re-examining the article, we realized that the sentence in question could be confusing to readers. To avoid any negative reading experiences, we have removed the sentence from the article.

Point 2: Ln 60: Some scholars have gained a better understanding of proteins’ functions in the proteome of ovarian carcinoma to gain a more comprehensive understanding, providing new avenues for developing more effective diagnostics and therapies [10,11].

This sentence is unclear.

Response 2: The wording of this sentence may be unclear upon further reading, so we have revised it to ensure that readers can understand its intended meaning: Scholars have gained a better understanding of the functions of proteins in the proteome of ovarian cancer and their impact on prognosis, providing new avenues for developing more effective diagnostics and therapies.

Point 3: Ln 179 The univariate Cox regression analyses of prognosis for differential proteins showed that they were significantly associated with outcome and could be a powerful indicator for the training set when using the proteome profiling data about ovarian cancer from the TCGA database.

Does this mean that the authors performed univariate Cox regression to identify differential proteins?

Response 3: You are absolutely right. Upon further reviewing the article, we found that the sentence was too verbose. Therefore, we have simplified it in the article while ensuring that it conveys the intended meaning. The original sentence is: The univariate Cox regression analyses of the prognostic relevance of differential proteins showed their significant association with the outcome, indicating their potential as a powerful indicator for the training set when using the proteome profiling data about ovarian cancer from the TCGA database. As a result, among these prognostic factors, 30 were found to be differentially expressed, with 18 having a suppressive effect (hazard ratio, HR<1) and 12 having a promotive effect (HR>1). And the modified sentence is: As a result of univariate Cox regression analyses, among these prognostic factors, 30 were found to be differentially expressed, with 18 having a suppressive effect (hazard ratio, HR<1) and 12 having a promotive effect (HR>1).

Point 4: Ln 189, it is unclear where the additional 20 proteins came from? What are prognosis factors? Do you mean factors that associate with or predict prognosis?

Response 4: Respected teacher, we are sorry for causing confusion. It was actually a typographical error that occurred during writing. After rechecking the data and to avoid unnecessary misunderstandings, we have decided to delete this sentence. Thank you again for your valuable suggestions, teacher.

Point 5: Ln 205 Is it the construction of a protein signature of prognosis?

Response 5: Indeed, that is correct. We constructed the ovarian cancer prognosis model based on proteins that were found to be closely associated with prognosis in our previous studies.

Point 6: Ln 235. This sentence is confusing.

Response 6: Thank you very much for your sincere feedback on the manuscript. We apologize for any confusion caused during your reading and have made the necessary revisions in the article.

Point 7: Ln 496 which study are you referring to? The phrase “The present study” makes it sound like the one the authors have submitted for review, but it is not in line with the text.

Response 7: Thank you for your valuable suggestion. You are right that our wording was confusing. Therefore, we have modified “The present study” to “A study” to avoid any misunderstanding.

Point 8: Ln 340 Prognosis-related proteins are a robust tool implemented in ovarian cancer for patient prognostication. Which proteins are these? Are they the signature discussed here?

 Response 8: We are sorry for the inconvenience caused to your reading experience. The sentence here indeed referred to the prognostic-related proteins obtained in the study. However, upon reviewing the article again, we found that this sentence was redundant and could be easily misunderstood, so we have deleted it.

Point 9: How do the protein models compare to transcriptomic based models?

 Response 9: Thank you very much for your valuable feedback on our article. Your question has greatly benefited us. Protein models and transcriptomic are two complementary relationships, with transcriptomic playing a regulatory role in protein expression as the starting point upstream of the central dogma, while proteins are the substances that play a role in gene expression, translation, modification, and other functions. Both can explain the occurrence, development, and prognosis of diseases. Perhaps the protein model is more closely related to the long-term prognosis of patients, but it must be regulated by transcriptomic. When used in clinical work to improve the outcome of ovarian cancer patients, we can consider both factors comprehensively.

Point 10: The term the authors use prognostic-related protein risk score is confusing. Perhaps it would be better to say protein-risk score only?

Response 10: Your suggestion is indeed very good and has prompted us to think deeply. After discussion, we have unanimously decided to adopt your suggestion to modify the description of “prognostic-related protein risk score” in the article to “protein-risk score”.

Point 11: Ln 488 furthermore is spelled wrong.

Response 11: I would like to express my gratitude for your valuable feedback. Regrettably, there was a spelling error in the article, and I have rectified it as per your suggestions.

Point 12: Ln 490 mutation is spelled wrong.

Response 12: I would like to express my gratitude for your valuable feedback. Regrettably, there was a spelling error in the article, and I have rectified it as per your suggestions.

Point 13: In the discussion how this would be implemented in the clinic could be discussed.

Response 13: Thank you very much for your modifications and suggestions on the article. Following your advice, we have added the following content to the discussion section: With the proposal of precision medicine, treatment targeting specific genes and targets has become a certain development trend. The treatment of the end product protein of gene translation will also provide a new direction for the treatment of ovarian cancer. Testing the expression of prognostic-related proteins in ovarian cancer patients can help predict their prognosis and timely follow-up. In case of disease recurrence or metastasis, early intervention can improve the prognosis and increase the survival rate.